# Generation of Conventional ^18^F-FDG PET Images from ^18^F-Florbetaben PET Images Using Generative Adversarial Network: A Preliminary Study Using ADNI Dataset

**DOI:** 10.3390/medicina59071281

**Published:** 2023-07-10

**Authors:** Hyung Jin Choi, Minjung Seo, Ahro Kim, Seol Hoon Park

**Affiliations:** 1Department of Nuclear Medicine, Ulsan University Hospital, Ulsan 44033, Republic of Korea; drchoihj@gmail.com; 2Department of Nuclear Medicine, Ulsan University Hospital, University of Ulsan College of Medicine, Ulsan 44033, Republic of Korea; 0733285@uuh.ulsan.kr; 3Department of Neurology, Ulsan University Hospital, University of Ulsan College of Medicine, Ulsan 44033, Republic of Korea; kahro@uuh.ulsan.kr

**Keywords:** Alzheimer, dementia, deep learning, GAN, FDG, florbetaben

## Abstract

*Background and Objectives*: ^18^F-fluorodeoxyglucose (FDG) positron emission tomography (PET) (PET_FDG_) image can visualize neuronal injury of the brain in Alzheimer’s disease. Early-phase amyloid PET image is reported to be similar to PET_FDG_ image. This study aimed to generate PET_FDG_ images from ^18^F-florbetaben PET (PET_FBB_) images using a generative adversarial network (GAN) and compare the generated PET_FDG_ (PET_GE-FDG_) with real PET_FDG_ (PET_RE-FDG_) images using the structural similarity index measure (SSIM) and the peak signal-to-noise ratio (PSNR). *Materials and Methods*: Using the Alzheimer’s Disease Neuroimaging Initiative (ADNI) database, 110 participants with both PET_FDG_ and PET_FBB_ images at baseline were included. The paired PET_FDG_ and PET_FBB_ images included six and four subset images, respectively. Each subset image had a 5 min acquisition time. These subsets were randomly sampled and divided into 249 paired PET_FDG_ and PET_FBB_ subset images for the training datasets and 95 paired subset images for the validation datasets during the deep-learning process. The deep learning model used in this study is composed of a GAN with a U-Net. The differences in the SSIM and PSNR values between the PET_GE-FDG_ and PET_RE-FDG_ images in the cycleGAN and pix2pix models were evaluated using the independent Student’s *t*-test. Statistical significance was set at *p* ≤ 0.05. *Results*: The participant demographics (age, sex, or diagnosis) showed no statistically significant differences between the training (82 participants) and validation (28 participants) groups. The mean SSIM between the PET_GE-FDG_ and PET_RE-FDG_ images was 0.768 ± 0.135 for the cycleGAN model and 0.745 ± 0.143 for the pix2pix model. The mean PSNR was 32.4 ± 9.5 and 30.7 ± 8.0. The PET_GE-FDG_ images of the cycleGAN model showed statistically higher mean SSIM than those of the pix2pix model (*p* < 0.001). The mean PSNR was also higher in the PET_GE-FDG_ images of the cycleGAN model than those of pix2pix model (*p* < 0.001). *Conclusions*: We generated PET_FDG_ images from PET_FBB_ images using deep learning. The cycleGAN model generated PET_GE-FDG_ images with a higher SSIM and PSNR values than the pix2pix model. Image-to-image translation using deep learning may be useful for generating PET_FDG_ images. These may provide additional information for the management of Alzheimer’s disease without extra image acquisition and the consequent increase in radiation exposure, inconvenience, or expenses.

## 1. Background

Alzheimer’s disease (AD) is the most common form of dementia and is characterized by progressive deterioration of memory and cognitive function. The characteristic neuropathological findings of AD consist of the accumulation of amyloid-β (Aβ) plaques in the extracellular space and the formation of neurofibrillary tangles in the intracellular space [1,2]. Early detection and assessment of abnormal Aβ deposition in the brain are important for proper management and treatment, as abnormal deposition of Aβ begins decades prior to the onset of cognitive decline [3].

Positron emission tomography (PET) imaging using ^18^F-florbetaben (FBB) (PET_FBB_) is a valuable tool for detecting Aβ in the brain and plays a vital role in the diagnosis and assessment of treatment response in AD. However, the use of PET_FBB_ alone is inadequate for differentiating AD from other forms of dementia, including Lewy body dementia [4]. Additionally, PET_FBB_ imaging may play a limited role in monitoring disease progression in AD cases with saturated Aβ deposition [5]. On the other hand, ^18^F-fluorodeoxyglucose (FDG) PET (PET_FDG_) imaging, which evaluates glucose metabolism, is useful for monitoring disease progression in AD [6] and differentiating AD from other types of dementia owing to the different patterns of glucose metabolism in the brain [7]. Therefore, the simultaneous use of PET_FDG_ and PET_FBB_ images may be synergistic and enhance the accuracy of AD diagnosis and enable a better assessment of disease progression. However, obtaining both PET_FBB_ and PET_FDG_ images in a patient poses significant challenges for practical reasons such as radiation exposure, inconvenience, and higher costs. Early-phase amyloid PET imaging reflects regional blood flow in the brain, and several studies have found that regional blood flow and glucose metabolism in the brain are coupled. Many studies have found similarities in tracer distribution between early-phase amyloid PET images and PET_FDG_ images, suggesting its value as an alternative imaging modality [8,9,10]. However, there are drawbacks, such as patient inconvenience, additional scan time, and limited availability of PET scanners, when compared to conventional amyloid PET imaging.

With the implementation of deep learning in medical imaging, image translation from one modality to another, such as PET_FBB_ images to magnetic resonance imaging (MRI) [11], early-phase 2β-carbomethoxy-3β-(4-iodophenyl)-N-(3-fluoropropyl) nortropane PET (^18^F-FP-CIT PET) images to PET_FDG_ images [12], has been widely conducted and evaluated in previous studies. Although a recent study reported on image-to-image translation using deep learning between amyloid tracers [13], there are limited studies focusing on generating PET_FDG_ images from PET_FBB_ images. The application of deep learning for generating PET_FDG_ images from PET_FBB_ images may be advantageous because it overcomes the challenges mentioned earlier.

Therefore, the objective of this study was to generate ^18^F-FDG PET (PET_GE-FDG_) images from ^18^F-florbetaben PET (PET_FBB_) images using a generative adversarial network (GAN) and compare PET_GE-FDG_ with real PET_FDG_ (PET_RE-FDG_) images using the structural similarity index measure (SSIM) and peak signal-to-noise ratio (PSNR).

## 2. Materials and Methods

### 2.1. Datasets

This study used the baseline PET_FBB_ image and PET_FDG_ image datasets downloaded from the Alzheimer’s Disease Neuroimaging Initiative (ADNI) database (adni.loni.usc.edu, accessed on 3 June 2022) [14]. The inclusion criterion for this study was the availability of both PET_FBB_ and PET_FDG_ images at baseline. Consequently, a total of 110 participants were included from the ADNI database (adni.loni.usc.edu, accessed 3 June 2022). The ADNI was launched in 2003 as a public–private partnership led by the principal investigator, Michael W. Weiner, MD, VA Medical Center and University of California, San Francisco. The primary objective of the ADNI was to test whether serial MRI, PET, and other biological markers can be combined with clinical and neuropsychological assessments to measure the progression of mild cognitive impairment and early AD. For the most up-to-date information, visit http://www.adni-info.org (accessed on 3 June 2022).

The baseline PET_FDG_ image acquisition was performed 30–60 min after injection of approximately 185 MBq of FDG, and image acquisition time was 30 min. On the other hand, the baseline PET_FBB_ image acquisition was performed 90–110 min after injection of approximately 300 MBq of FBB and image acquisition time was 20 min. The PET_FBB_ and PET_FDG_ images consisted of four and six subsets, respectively. Each subset was acquired every 5 min during the image acquisition process. All subsets of paired PET_FBB_ and PET_FDG_ images were randomly sampled and divided into 249 and 95 subsets for training and validation, respectively. The Institutional Ethics Committee of the Ulsan University Hospital reviewed this observational study and waived the requirement for informed consent (IRB file number: UUH2022-05-028).

### 2.2. Deep-Learning Model with Image Preprocessing

Because all the PET images had different matrix sizes and our computer resources were limited, all images were resampled with a matrix size of 64 (height) × 64 (width) × 1 (color channel). A total of 15,936 two-dimensional (2D) images were prepared for training, and 6080 were used for validation. A “Bit Stored” attribute in the Digital Imaging and Communications in Medicine (DICOM) header of each PET image was used to determine the divisor for data rescaling, which converted unsigned integer pixel values to floating point values (range: 0.0–1.0).

This study adopted unsupervised image-to-image translation models using GAN and U-Net architecture that were based on “CycleGAN and pix2pix in PyTorch” (https://github.com/junyanz/pytorch-CycleGAN-and-pix2pix, accessed on 10 March 2023) [15,16,17]. The GAN developed for the purpose of translating unpaired images consisted of a generator and a discriminator. The generator was responsible for generating output images based on input images. In the generator, the key network used for extracting features from the input images and delineating the output images was U-Net. The max-pooling process used in the original U-Net was omitted to improve training efficiency. Additionally, leaky RELU was used for a downward activation function instead of RELU, which was the activation function in the original U-NET. As for the discriminator, only the left half of the U-NET architecture was used because the right half was the part of the image generation that was not necessary for discrimination. In other words, the discriminator was operated by using the extracted features of the images and the mean squared error for a loss function that calculated differences between the generated and ground truth target images. The architectural designs of the generator and the discriminator are illustrated in Figure 1 and Figure 2. The architecture of the cycleGAN model is shown in Figure 3. The model architecture is expressed as follows.
*G*_*A*(*A*) → *B*(1)
*G*_*B*(*B*) → *A*
(2)
*D*_*A*(*G*_*A*(*A*)) → *B*
(3)
*D*_*B*(*G*_*B*(*B*)) → *A*
(4)
*Loss*(*A*, *B*) = *D*_*A*(*G*_*A*(*A*)) + *D*_*B*(*G*_*B*(*B*)) + |*G*_*B*(*G*_*A*(*A*)) − *A*| + |*G*_*A*(*G*_*B*(*B*)) − *B*| + |*G*_*A*(*B*) − *B*| + |*G*_*B*(*A*) − *A*|(5)

A and B represent PET_FBB_ and PET_FDG_ images, respectively. The right-hand variables of the arrows indicate the ground-truth target images. G_A (1) and G_B (2) are the same generators; however, G_A (1) is a forward generator that creates B from A, and G_B (2) is a backward generator that creates A from B. D_A (3) and D_B (4) are the same discriminators that determine the differences between the real and forward-/backward-generated images. Loss (A, B) (5) represents a loss function that addresses the differences between the generated and ground-truth target images. The cycleGAN and pix2pix models were then trained to minimize loss (A, B). The cycleGAN model uses all paired generators (1,2) and discriminators (3,4) mentioned above (Figure 3). In contrast, the pix2pix model consists of only a forward generator (1) and a discriminator (3) that was drawn in Figure 4. Minor modifications were made to the original Python code to allow it to run on Python 3.10, PyTorch 2.0.0, CUDA 11.7, and Windows 10.

### 2.3. Statistical Analysis

Participant demographics between the training and validation datasets were compared using the independent Student’s *t*-test and Mann–Whitney U test for continuous and categorical variables, respectively. The similarity between the PET_GE-FDG_ and PET_RE-FDG_ images was determined using SSIM [18]. The formula for the SSIM is as follows:$$\mathrm{SSIM}(x,\;y)=\frac{\left(2\mu_{x}\mu_{y}+C_{1})(2\sigma_{xy}+C_{2}\right)}{\left(\mu_{x}^{2}+\mu_{y}^{2}+C_{1})(\sigma_{x}^{2}+\sigma_{y}^{2}+C_{2}\right)}$$
where x and y are PET_RE-FDG_ and PET_GE-FDG_ images, respectively, to be compared, and μ and σ are the mean and standard deviation of these images. The pixel value ranges of these images were used to calculate constant variables C_1_ and C_2_, which were used to stabilize the division with a weak denominator. When a greater similarity exists between the x and y images, the SSIM value approaches 1. A greater anticorrelation between these images resulted in an SSIM value closer to −1. The SSIM value was close to zero when no similarity was observed between the images. The PSNR between the images was also measured. The PSNR is computed using the following formula:$$\mathrm{Mean}\;\mathrm{squared}\;\mathrm{error}\;(\mathrm{MSE})=\frac{1}{mn}\sum_{i=0}^{m-1}\sum_{j=0}^{n-1}{\left[I\left(i,j\right)-K\left(i,j\right)\right]}^{2}$$
PSNR (dB) = 20log_10_(MAX_I_) − 10log_10_ (MSE)
where ‘*m*’ and ‘*n*’ represent the width and height of an image, respectively, and ‘*K*’ represents the noisy approximation of the image. The term ‘MAX_I_’ refers to the maximum possible pixel value of the image. A higher PSNR value indicated that the images were more similar.

An independent Student’s *t*-test was performed to compare the SSIM values of the image datasets using the cycleGAN and pix2pix models. Statistical significance was set as *p* ≤ 0.05.

## 3. Results

### 3.1. Baseline Demographics

Of the 110 participants, 82 (75%) were in the training group and 28 (25%) were in the validation group. There were no significant differences in age, sex, or diagnosis between the training and validation groups (*p*-values for age, sex, and diagnosis were 0.68, 0.72, and 0.55, respectively). Table 1 presents the detailed demographics of the participants.

### 3.2. Differences in SSIM and PSNR Values between PET_RE-FDG_ and PET_GE-FDG_ Images

PET_GE-FDG_ images were created using the cycleGAN and pix2pix models with training times of 62 h 36 min and 21 h 22 min, respectively. The mean SSIM (SSIM_mean_) between the PET_RE-FDG_ and PET_GE-FDG_ images was 0.768 for the cycleGAN model and 0.745 for the pix2pix model (Table 2). The cycleGAN model showed significantly higher SSIM values than the pix2pix model (*p* < 0.001). The SSIM values in the cycleGAN and pix2pix models are represented by the box plots in Figure 5. The mean PSNR (PSNR_mean_) between the PET_RE-FDG_ and PET_GE-FDG_ images was 32.4 for the cycleGAN and 30.7 for the pix2pix models (Table 3). The PSNR values of the cycleGAN model were significantly higher than those of the pix2pix model (*p* < 0.001). The PSNR values for the cycleGAN and pix2pix models are shown in the box plots provided in Figure 6. Representative PET_GE-FDG_ and PET_RE-FDG_ images from one participant are shown in Figure 7 (A, PET_FBB_; B, PET_GE-FDG_ using the pix2pix model; C, PET_GE-FDG_ using the cycleGAN model; and D, PET_RE-FDG_, from left to right).

## 4. Discussion

With an increasingly aging society, the incidence of neurodegenerative disorders may also increase, particularly AD, which is the most common form of dementia [19]. The early and accurate diagnosis of AD is important for the medical and socioeconomic care of patients [20]. It allows for the early and appropriate management of AD patients, with the medical faculty focusing on preserving cognitive function and preventing irreversible damage [21]. For this reason, there has been a considerable number of studies dedicated to utilizing non-invasive imaging modalities, such as amyloid PET and PET_FDG_, in recent years. There has been increasing interest in the potential of early amyloid PET image as an alternative to PET_FDG_. As in our study, generating PET_GE-FDG_ images from PET_FBB_ images using deep learning has clinical implications in terms of reducing the cost and radiation exposure of the patient and eliminating the inconvenience of repeat examinations.

The neuropathological hallmarks of AD are the presence of intracellular neurofibrillary tangles and extracellular amyloid plaques [22]. Increased amyloid deposition in the brain is known to be associated with cognitive decline, and these deposits have been detected in AD patients approximately 10–15 years prior to symptom onset. Since PiB was first used in research, the only FDA-approved clinical amyloid PET tracers to date are FBB, ^18^F-Florbetapir, and ^18^F-Flutemetamol, an imaging test that allows for the visual assessment of abnormal amyloid deposition in the brain [23]. Although amyloid PET imaging is highly specific for assessing the amyloid burden in the brain, there are difficulties in assessing the progression of AD in patients who exhibit high levels of amyloid deposition at the time of diagnosis [24]. In addition, positive findings of amyloid deposition can be seen not only in AD but also in other types of dementia, such as Lewy body dementia [25].

The brain utilizes approximately a quarter of the body’s glucose on a daily basis. Glucose is transported from the blood to the brain cells via glucose transporters. FDG, which is a glucose analog, is transported to brain cells via the same pathway but undergoes phosphorylation within the cell, which prevents it from being released from the cell. FDG accumulates without further glycolysis and is a good reflection of glucose uptake in brain cells [7]. The stimulation of neurons has been reported to coincide with FDG uptake at neuronal terminals, indicating that FDG uptake in the brain reflects neuronal activity [7]. Thus, PET_FDG_ imaging serves as a functional imaging biomarker for assessing regional brain dysfunction caused by neuronal injury in AD. AD is characterized by decreased glucose metabolism in the posterior cingulate cortex, precuneus, and parieto–temporal cortex on the PET_FDG_ image, and in advanced cases, the decreased FDG uptake may extend to the frontal cortex. It is a non-invasive imaging test that is useful for the evaluation of disease extent in AD as well as for the differential diagnosis of other types of dementia, in which cases amyloid PET imaging may have a limited role [5,26,27,28,29]. It can also be used to predict the progression from mild cognitive impairment to AD [30] and to classify subtypes of AD [31]. However, decreased FDG uptake in the brain on PET_FDG_ imaging is indicative of neurodegeneration. Thus, PET_FDG_ imaging is not an appropriate imaging test for the early diagnosis of AD [32].

Early-phase ^11^C-Pittsburgh compound B (PiB) PET (PET_PiB_) imaging, which is obtained within the first few minutes after PiB injection, reflects the cerebral blood pool due to the lipophilic nature and high extraction fraction of PiB [33]. The close relationship between blood supply and glucose consumption in the brain has been well-documented in several studies. In regions of the brain with neuronal injury, glucose hypometabolism and hypoperfusion are often concurrent [34,35]. Decreased tracer uptake on early-phase PET_PiB_ images is reported to closely correlate with hypometabolism in PET_FDG_ images and low mini-mental state examination scores in patients with early-stage AD. Thus, both abnormal amyloid deposition and neurodegeneration extent in the brain may be assessed with a conventional PET_PiB_ image [8,9]. Another amyloid tracer, ^18^F-florbetapir, has also been reported to have a strong correlation with PET_FDG_ in early-phase ^18^F-florbetapir PET imaging [36]. In recent studies regarding the early-phase PET_FBB_ image, the early-phase PET_FBB_ image showed a close correlation with the PET_FDG_ image [25,37,38]. In one study, early-phase PET_FBB_ images showed a slightly stronger correlation than PET_PiB_ images, suggesting that obtaining PET_FBB_ images at dual time points with a single radioisotope injection has the advantages of allowing for both accurate diagnosis and assessment of progression in patients with AD [10]. This suggests that early-phase amyloid PET image could serve as a clinically viable alternative to the PET_FDG_ image for assessing neuronal injury in patients with dementia.

There has been increasing interest in using artificial intelligence (AI) to accurately assess cognitive function in patients with AD. This is because AI has the potential to overcome the diagnostic limitations of existing molecular biomarkers (such as amyloid plaque and tau in cerebrospinal fluid) and imaging methods (such as computed tomography (CT), MRI, amyloid PET imaging, and PET_FDG_ imaging). AI can also help analyze and interpret complex and large amounts of information from the brain. Most AI-based research in AD focuses on developing AI algorithms for the classification or diagnosis of AD and for developing biomarkers for the early detection of AD [39]. The current study focused on image-to-image translation using deep learning to decrease the clinical burden associated with obtaining multimodality imaging. We aimed to investigate whether AI can generate PET_GE_-_FDG_ image from conventional amyloid PET image. Image transformation using deep learning in medical imaging has been widely studied [40,41]. Most of the studies have focused on image translation between MRI and CT images, while some have studied image translation between conventional radiologic imaging (CT or MRI) and PET images [41]. A recent study reported on image-to-image translation using deep learning between amyloid tracers [13]. In one of our previous studies, we reported on image-to-image translation between the ^18^F-FP-CIT PET image and the PET_FDG_ image using deep learning. The study results revealed that the early-phase ^18^F-FP-CIT PET image that was generated was significantly similar to the PET_FDG_ image [12]. The generation of PET_FBB_ from PET_FDG_ using deep learning has also been reported [42]. To the best of our knowledge, no study has attempted to generate PET_FDG_ images from PET_FBB_ images. This is a preliminary study that shows the potential of using two deep-learning models (cycleGAN and pix2pix) for generating PET_FDG_ images from conventional PET_FBB_ images. PET_GE-FDG_ images may benefit patients by further reducing examination time (by acquiring only a single time point conventional PET_FBB_ image instead of dual-time point PET_FBB_ image). The generation of PET_FDG_ images from PET_FBB_ images might be challenging because of the negative correlation between the regional uptakes in PET_FDG_ and PET_FBB_ images [43]. Additionally, PET_FBB_ images generally have a lower image quality than PET_FDG_ images, especially when there is a negative amyloid burden.

SSIM and PSNR were used to compare PET_GE-FDG_ and PET_RE-FDG_. SSIM was developed to predict the perceived quality of digital images and measure the similarity between two images [18]. Since then, SSIM has been widely used for image comparison by detecting perceived structural changes during image processing. PSNR is a quantitative measure of image denoising quality. SSIM aligns more closely with the human visual perception of image quality when compared with PSNR [44]. SSIM and PSNR values of the cycleGAN model was higher than those of the pix2pix model in this study. Although higher SSIM and PSNR were not always equal to higher visual quality, PET_GE-FDG_ using the cycleGAN model was statistically closer to the PET_RE-FDG_ image than the PET_GE-FDG_ image using the pix2pix model.

Two different GAN models, cycleGAN and pix2pix, were used to generate PET_FDG_ images from PET_FBB_ in this study. Compared with the SSIM values of the PET_GE-FDG_ images using the cycleGAN model, the lower SSIM values of the PET_GE-FDG_ images using the pix2pix model may be related to the misalignment between the PET_GE-FDG_ and PET_RE-FDG_ images. The pix2pix model is specialized for paired image-to-image translation. Therefore, the contours of the PET_FBB_ and PET_FDG_ images must be aligned to correct the misalignment before training the pix2pix model [11]. The correction process for the misalignment is time-consuming and labor-intensive. In contrast, the cycleGAN model represents unpaired image-to-image translation, and no preprocessing is required for image alignment. In this respect, the cycleGAN model for the generation of PET_FDG_ from PET_FBB_ may be considered more appropriate than the pix2pix model.

This study had some limitations. First, the numbers of training and validation datasets were small. However, this limited dataset size is reflective of the reality of medical practice, where it is difficult to perform both tests (PET_FBB_ and PET_FDG_ images). In other words, medical imaging data are often unpaired. In this regard, the application of deep learning with cycleGAN model can have a significant clinical impact on the accurate assessment of AD. Second, the size of each image was small. These limitations might affect the image quality, although image size reduction is inevitable because of lower computational resources [45]. Therefore, the image quality of PET_GE-FDG_ is insufficient for visual assessments in daily practice. Nevertheless, data augmentation techniques, including patching, flipping, and resizing, have been used to overcome these limitations. Subsequently, PET_GE-FDG_ images with tolerable SSIM were generated. A larger number of datasets with relevant image sizes should be made available to improve the visual quality of PET_GE-FDG_ images in further studies.

To the best of our knowledge, this study represents the first attempt to evaluate whether images close to PET_FDG_ images may be generated from conventional PET_FBB_ images using deep learning. Despite the aforementioned drawbacks and limitations, our study results suggest that deep learning may reduce the time, cost, and patient inconvenience of additional early-phase scanning in PET_FBB_ images and provide information on the regional glucose metabolism of the brain at the same time. Future studies with larger sample sizes are warranted to evaluate the correlation of PET_GE-FDG_ and PET_RE-FDG_ images, which might provide valuable clinical evidence in this field.

## 5. Conclusions

We generated PET_FDG_ from PET_FBB_ images using the cycleGAN and pix2pix models. The cycleGAN model generated PET_FDG_ images with significantly higher SSIM and PSNR than the pix2pix model. We demonstrated that PET_GE-FDG_ image using cycleGAN may have an image quality and similarity closer to PET_RE-FDG_ image and help provide proper management of AD by minimizing additional radiation risk and inconvenience caused to the patient by extra image acquisition such as early-phase amyloid PET image.

## Figures and Tables

**Figure 1 medicina-59-01281-f001:**
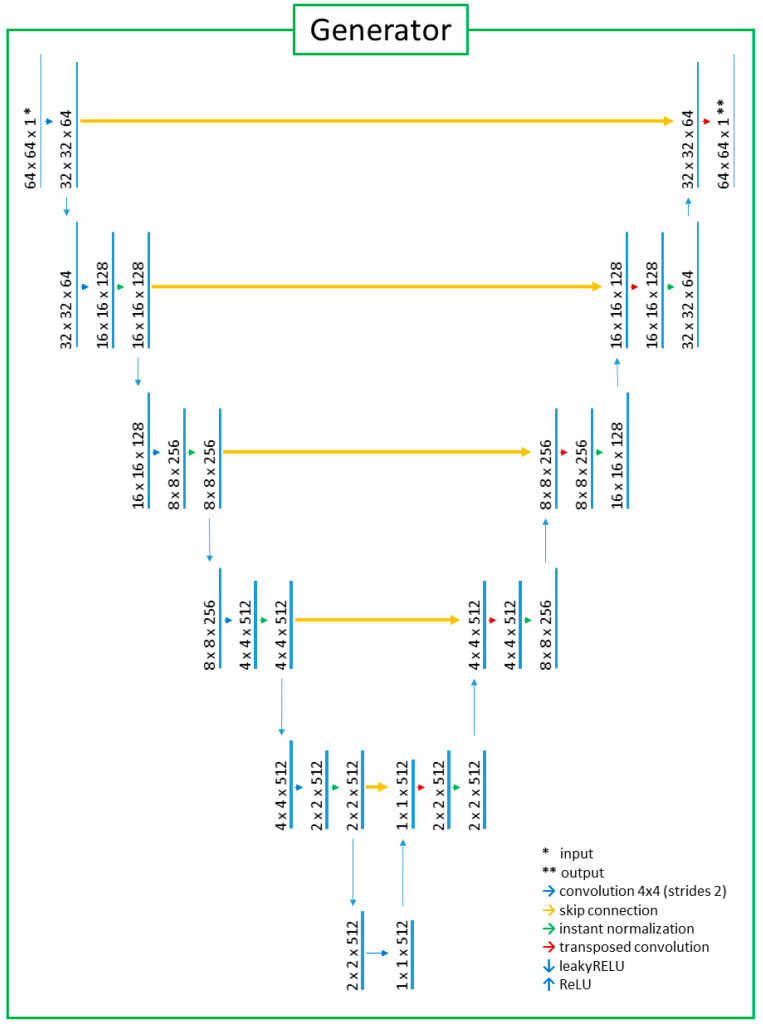
The generator with the modified U-Net architecture. The input/output image format was 64 (width) × 64 (height) × 1 (channel). After the vertical median line was virtually drawn on the U-Net diagram, the left half-side of the U-Net was used for feature extraction from input images. The other right half-side was used for image generation.

**Figure 2 medicina-59-01281-f002:**
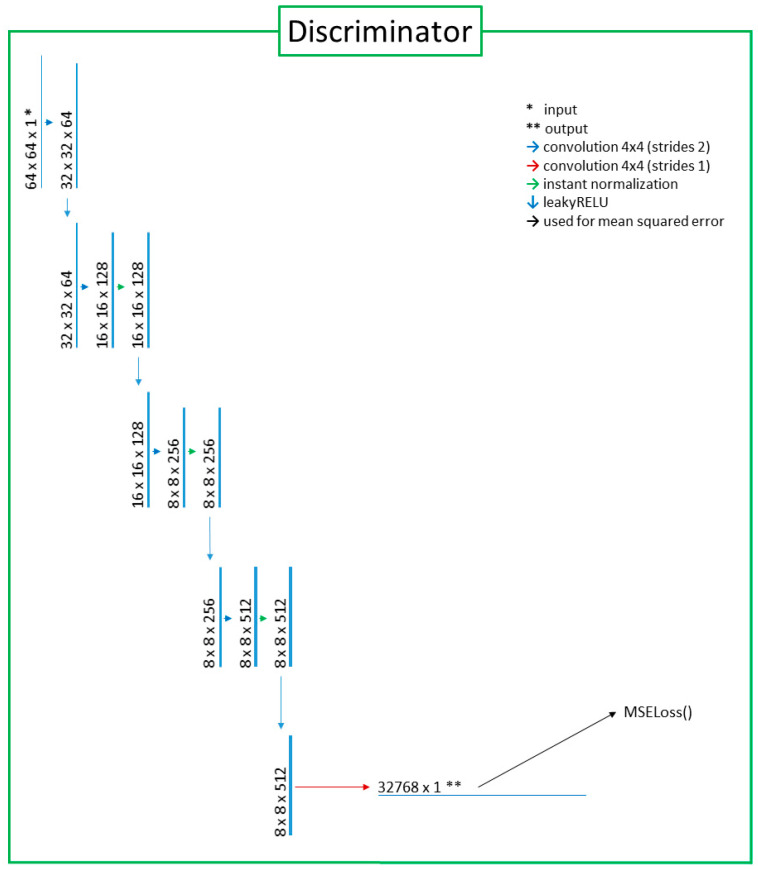
The discriminator with the left half-side of the U-Net architecture. Features extracted from images were fed to the mean squared error (MSE) loss function for comparison.

**Figure 3 medicina-59-01281-f003:**
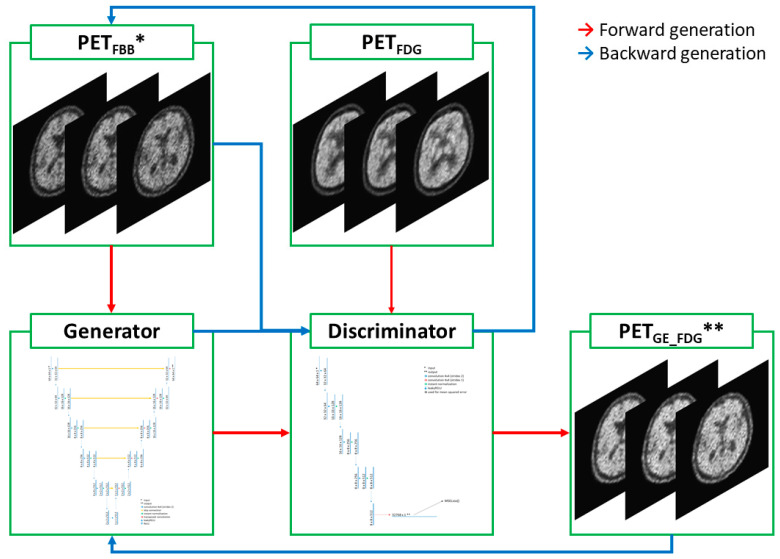
Diagram of the cycleGAN model. In forward generation, * PET_FBB_ images were put into the generator and ** PET_GE-FDG_ images were generated. The discriminator compared these PET_GE-FDG_ images with ground-truth PET_FDG_ images. In backward generation, the PET_FBB_ images were generated from PET_GE-FDG_ images using the same generator and discriminator.

**Figure 4 medicina-59-01281-f004:**
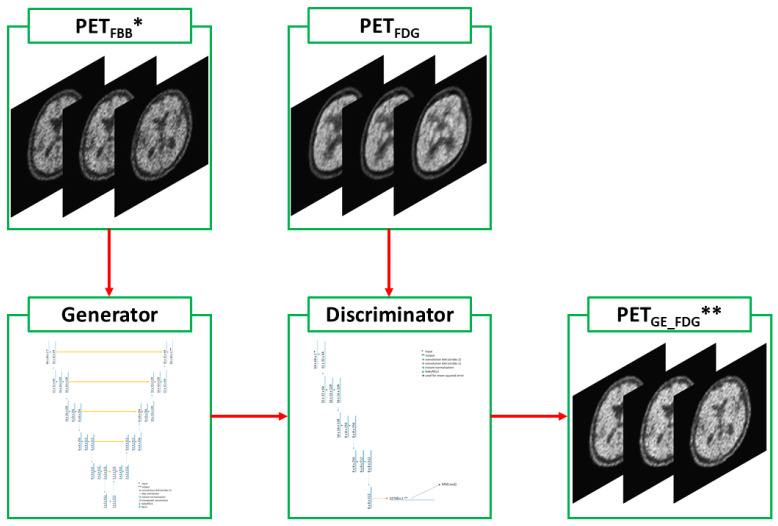
Diagram of the pix2pix model. ** PET_GE-FDG_ images were generated from * PET_FBB_ images using the one-way process, the same forward generation of the cycleGAN model.

**Figure 5 medicina-59-01281-f005:**
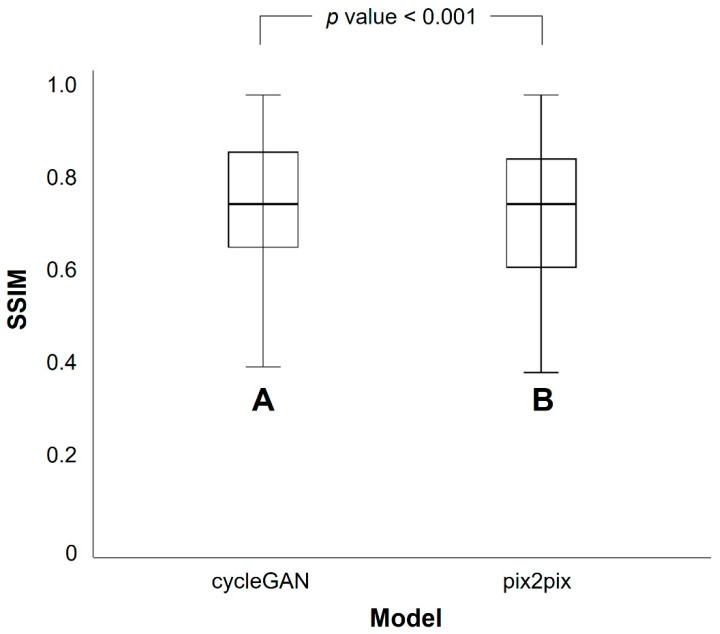
Difference in mean structural similarity index measure (SSIM) values between the cycleGAN (A) and pix2pix (B) models. The cycleGAN model shows a significantly higher SSIM than the pix2pix model.

**Figure 6 medicina-59-01281-f006:**
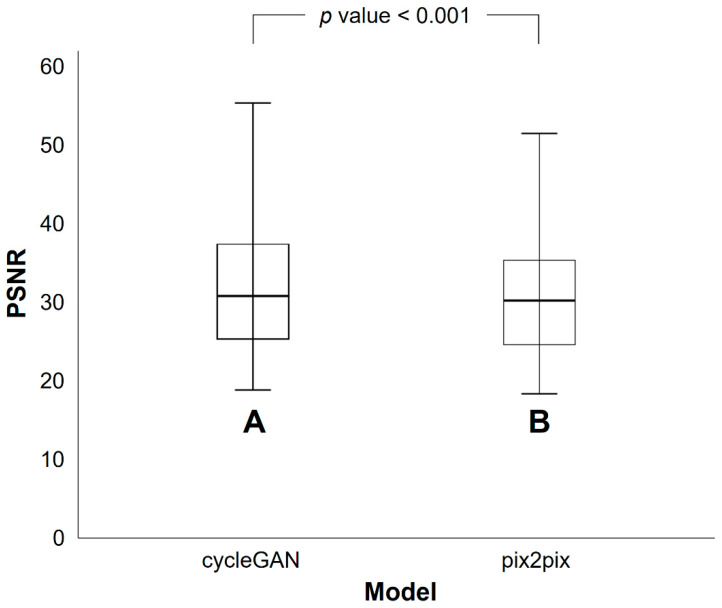
Difference in mean peak signal-to-noise ratio (PSNR) values between the cycleGAN (A) and pix2pix (B) models. The cycleGAN model shows a significantly higher PSNR than the pix2pix model.

**Figure 7 medicina-59-01281-f007:**
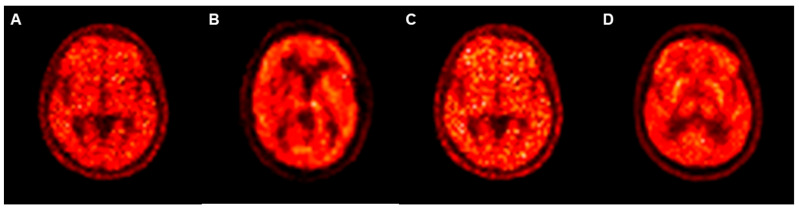
A representative case of PET_GE-FDG_ and PET_RE-FDG_ images. PET_FBB_ (**A**), PET_GE-FDG_ using the pix2pix model (**B**), PET_GE-FDG_ using the cycleGAN model (**C**), and PET_RE-FDG_ (**D**) are arranged from left to right.

**Table 1 medicina-59-01281-t001:** Baseline participant demographics.

	Training Group	Validation Group	Total
Number	82 (75%)	28 (25%)	110
Age *, years	72.8 ± 7.8	72.0 ± 9.1	
Sex *, n (%)			
Male	50 (61%)	16 (57%)	66 (60%)
Female	32 (39%)	12 (43%)	44 (40%)
Diagnosis ^†^, n (%)			
Normal	1 (1%)	1 (4%)	2 (2%)
MCI	60 (73%)	21 (75%)	81 (74%)
AD	21 (26%)	6 (21%)	27 (24%)

Abbreviations: MCI, mild cognitive impairment; AD, Alzheimer’s disease. The *p* values for age, sex, and diagnosis were 0.68, 0.72, and 0.55, respectively, * Independent Student’s *t*-test (*p* > 0.05) indicates no statistical significance. ^†^ Mann–Whitney U test (*p* > 0.05) indicates no statistical significance.

**Table 2 medicina-59-01281-t002:** Mean SSIM values between cycleGAN and pix2pix models.

	CycleGAN Model	Pix2pix Model	*p* Value *
Mean	0.768	0.745	<0.001
Standard deviation	0.135	0.143	

Abbreviations: SSIM, structural similarity index measure; GAN, generative adversarial network. * Independent *t*-test (*p* < 0.05) indicates statistical significance.

**Table 3 medicina-59-01281-t003:** Mean PSNR values between cycleGAN and pix2pix models.

	CycleGAN Model	Pix2pix Model	*p* Value *
Mean	32.4	30.7	<0.001
Standard deviation	9.5	8.0	

Abbreviations: PSNR, peak signal-to-noise ratio; GAN, generative adversarial network. * Independent *t*-test (*p* < 0.05) indicates statistical significance.

## Data Availability

Not applicable.

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
