# Peer review of "Generation of Conventional 18F-FDG PET Images from 18F-Florbetaben PET Images Using Generative Adversarial Network: A Preliminary Study Using ADNI Dataset"

_medicina, 2023, doi:10.3390/medicina59071281_

Round 1
Reviewer 1 Report
The paper titled "Generation of 18F-FDG PET Images from Delayed-phase 18F-florbetaben PET Images Using Generative Adversarial Network" explores the generation of PET FDG-like images from PET RE_FBB images through the application of deep learning. While the study presents potential results, the reviewer feels its acceptance in its current stage is not recommended due to several significant shortcomings. Firstly, the paper suffers from numerous linguistic errors that severely impact its clarity and readability. These errors should be addressed to enhance the overall quality of the manuscript. Additionally, the literature review is inadequate, lacking a comprehensive analysis of existing works in the field. The authors have failed to reference relevant papers on the generation of PET FDG from PET FBB using deep learning methods. This omission is particularly noticeable in the absence of references to papers related to the proposed deep learning model.Furthermore, the paper lacks methodological advancements or novel approaches. There is a lack of significant contributions to the existing body of knowledge. Additionally, the manuscript fails to compare the proposed method with similar approaches, which is crucial for evaluating its effectiveness and superiority. Moreover, the figures in the manuscript suffer from unclear descriptions in the legends.
Lastly, the obtained results, while promising, are not sufficiently significant for publication in this journal. The authors should provide a detailed explanation of the additional information gained using the proposed method for the management of Alzheimer's disease. This would demonstrate the practical relevance and impact of the research.
Extensive editing of English language required
Author Response
Dear Reviewer
Thank you for your thoughtful comments and suggestions.
We have checked and revised the manuscript as you have suggested.
The changes can be tracked in the manuscript.
For your convenience, the major changes are also attached below.
We would also like to point out that there are two corresponding authors to this article.
Thank you very much.
Sincerely,
Seol Hoon Park
================================================================
REVIEWER 1
The paper titled "Generation of 18F-FDG PET Images from Delayed-phase 18F-florbetaben PET Images Using Generative Adversarial Network" explores the generation of PET FDG-like images from PET RE_FBB images through the application of deep learning. While the study presents potential results, the reviewer feels its acceptance in its current stage is not recommended due to several significant shortcomings.
Point 1. Firstly, the paper suffers from numerous linguistic errors that severely impact its clarity and readability. These errors should be addressed to enhance the overall quality of the manuscript. Additionally, the literature review is inadequate, lacking a comprehensive analysis of existing works in the field. The authors have failed to reference relevant papers on the generation of PET FDG from PET FBB using deep learning methods. This omission is particularly noticeable in the absence of references to papers related to the proposed deep learning model.
- After reflecting on the points made, we have corrected linguistic errors to improve the clarity and readability of the manuscript, and added additional information and references to studies using deep learning in the PET imaging. To our knowledge, there are no studies that generate FDG like early phase FBB PET from conventional FBB PET, so we believe our preliminary study is meaningful, albeit with some limitations.
- Page 10 , Discussion section 5 paragraph
There has been increasing interest in using artificial intelligence (AI) to accurately assess cognitive function in patients with AD. This is because AI has the potential to overcome the diagnostic limitations of existing molecular biomarkers (such as amyloid plaque and tau in cerebrospinal fluid) and imaging methods (such as computed tomography (CT), MRI, amyloid PET imaging, and PETFDG imaging). AI can also help analyze and interpret complex and large amounts of information from the brain. Most AI-based research in AD focuses on developing AI algorithms for classification or diagnosis of AD and for developing biomarkers for early detection of AD [39]. The current study focused on image-to-image translation using deep learning to decrease the clinical burden associated with obtaining multimodality imaging. We aimed to investigate whether AI can generate PETFDG-like early phase amyloid image from conventional amyloid PET image. Image transformation using the deep learning in medical imaging has been widely studied [40-42]. Most of the studies have focused on image translation between MRI and CT images, while some have studied image translation between conventional radiologic imaging (CT or MRI) and PET images [42]. A recent study reported on image-to-image translation using deep learning between amyloid tracers [12]. In one of our previous studies, we reported on image-to-image translation between 18F-FP-CIT PET image and PETFDG image using deep learning. The study results revealed that the early phase 18F-FP-CIT PET image that was generated was significantly similar to PETFDG image [11]. The generation of PETFBB from PETFDG using deep learning has also been reported [43]. To the best of our knowledge, no study has attempted to generate PETFDG images from PETFBB images. This is a preliminary study that shows the potential of using two deep learning models (cycleGAN and pix2pix) for generating PETFDG-like images from conventional PETFBB images. PETGE_FDG images may benefit patients by further reducing examination time (by acquiring only single time point conventional PETFBB image instead of dual-time point PETFBB image). The generation of PETFDG images from PETFBB images might challenging because of the negative correlation between the regional uptakes in PETFDG and PETFBB images [44]. Additionally, PETFBB images generally have a lower image quality than PETFDG images, especially when there is a negative amyloid burden.
Point 2. Furthermore, the paper lacks methodological advancements or novel approaches. There is a lack of significant contributions to the existing body of knowledge. Additionally, the manuscript fails to compare the proposed method with similar approaches, which is crucial for evaluating its effectiveness and superiority. Moreover, the figures in the manuscript suffer from unclear descriptions in the legends.
- The purpose of our paper was not to compare and demonstrate superiority of methods, but to evaluate the feasibility of image to image translation in a manner similar to deep learning studies performed on conventional PET imaging. we wrote a more detailed description of our methodology and added figures in the revised manuscript.
Page 6, Materials and Methods, Deep learning model with image preprocessing, 2nd paragraph
The GAN developed for the purpose of translating the unpaired images consisted of a generator and a discriminator. The generator was responsible for generating output images based on input images. In the generator, the key network for extracting features from the input images and delineating the output images was U-Net. The max-pooling process used in the original U-Net was omitted to improve training efficiency. Additionally, leaky RELU was used for a down-ward activation function instead of RELU, which was the activation function in the original U-NET. As for the discriminator, only the left half of the U-NET architecture was used because the right half was the part of the image generation that was not necessary for discrimination. In other words, the discriminator was operated by the extracted features of the images and the mean squared error for a loss function that calculated differences between the generated and ground truth target images. The architectural designs of the generator and the discriminator are illustrated in Figures 1 and 2. The architecture of the cycleGAN model is shown in Figure 3.
Figure Legends
Figure 1. The generator with the modified U-Net architecture. The input/output image format was 64 (width) × 64 (height) × 1 (channel). After the vertical median line was virtually drawn on the U-Net diagram, the left half-side of the U-Net was used for feature extraction from input images. The other right half-side was used for image generation.
Figure 2. The discriminator with the left half-side of the U-Net architecture. Features extracted from images were fed to the mean squared error (MSE) loss function for comparison.
Figure 3. Diagram of the cycleGAN model. In forward generation, *PETFBB images were put into the generator and **PETGE_FDG images were generated. The discriminator compared these PETGE_FDG images with ground-truth PETFDG images. In backward generation, the PETFBB images were generated from PETGE_FDG images using the same generator and discriminator.
Figure 4. Diagram of the pix2pix model. **PETGE_FDG images were generated from *PETFBB images using the one-way process, the same forward generation of the cycleGAN model.
Point 3. Lastly, the obtained results, while promising, are not sufficiently significant for publication in this journal. The authors should provide a detailed explanation of the additional information gained using the proposed method for the management of Alzheimer's disease. This would demonstrate the practical relevance and impact of the research.
- Our study was based on patients with both FDG PET and FBB PET in the ADNI dataset, so it suffers from small numbers, which makes it difficult to say that the study itself can actually provide clinicians with follow-up information on AD. However, our study shows promise that images similar to early phase FBB PET images can be created using deep learning, and we believe this study needs to be validated with a larger patient population.

Reviewer 2 Report
1. Although the term “delayed-phase 18F-florbetaben PET (PETFBB) images” is mentioned in the title and conclusions, the article does not provide any explanation or discussion of this term.I don't think there is enough evidence to say “This study demonstrated the generation of PETGE-FDG from PET-FBB using two DL models . We also demonstrated that this generation is possible even when using delayed images from PET-FBB.” in the conclusions.
2. There is no U-Net structure in the cycleGAN,but “The deep learning model used in this study is composed of a GAN with a U-Net.”is written in the Abstract.
3. What’s the purpose of Generation of 18F-FDG PET Images from Delayed-phase 18F-florbetaben PET Images Using Generative Adversarial Network, Is it intended to help doctors with diagnosis in reality or for computer-aided diagnosis? For computer-aided diagnosis, SSIM as a criterion for assessing the quality of the generated images is inadequate: SSIM places more emphasis on the evaluation of structural information and perceptual quality and is suitable for scenarios that require human eye observation. Using both SSIM and PSNR would make the conclusion more convincing.
4. I could not find a paragraph in reference [5] that matches the meaning of the quoted sentence.
5. The tracers used in reference [8,9] and reference [10] are PIB and florbetapir, respectively,which are not equivalent to florbetaben.
6. It would be clearer to describe image-generation model architecture and the structure of CycleGAN/pix2pix models in diagrams instead of unclear text description.
7. The article does not mention the connection between the two modalities and the advantages and disadvantages of applying each to diagnosis.
Good
Author Response
Dear Reviewer
Thank you for your thoughtful comments and suggestions.
We have checked and revised the manuscript as you have suggested.
The changes can be tracked in the manuscript.
For your convenience, the major changes are also attached below.
We would also like to point out that there are two corresponding authors to this article.
Thank you very much.
Sincerely,
REVIEWER 2
- Although the term “delayed-phase 18F-florbetaben PET (PETFBB) images” is mentioned in the title and conclusions, the article does not provide any explanation or discussion of this term. I don't think there is enough evidence to say “This study demonstrated the generation of PETGE-FDG from PET-FBB using two DL models. We also demonstrated that this generation is possible even when using delayed images from PET-FBB.” in the conclusions.
- The term “delayed-phase FBB PET” was used to emphasize that we did not use early phase images of a dynamic FBB PET, as others tried before. We used FBB PET images obtained 90 minutes after radiotracer injection. We agree with you and think that “delayed-phase” may cause confusion for the readers. We now either erased “delayed-phase” from the manuscript or changed it with the word “conventional”.
- There is no U-Net structure in the cycleGAN, but “The deep learning model used in this study is composed of a GAN with a U-Net.”is written in the Abstract.
- We added a paragraph concerning the U-net structure to the manuscript.
Page 6, Methods and Methods, Deep learning model with image preprocessing, 2nd paragraph
The GAN developed for the purpose of translating the unpaired images consisted of a generator and a discriminator. The generator was responsible for generating output images based on input images. In the generator, the key network for extracting features from the input images and delineating the output images was U-Net. The max-pooling process used in the original U-Net was omitted to improve training efficiency. Additionally, leaky RELU was used for a down-ward activation function instead of RELU, which was the activation function in the original U-NET. As for the discriminator, only the left half of the U-NET architecture was used because the right half was the part of the image generation that was not necessary for discrimination. In other words, the discriminator was operated by the extracted features of the images and the mean squared error for a loss function that calculated differences between the generated and ground truth target images. The architectural designs of the generator and the discriminator are illustrated in Figures 1 and 2. The architecture of the cycleGAN model is shown in Figure 3.
Figure Legends
Figure 1. The generator with the modified U-Net architecture. The input/output image format was 64 (width) × 64 (height) × 1 (channel). After the vertical median line was virtually drawn on the U-Net diagram, the left half-side of the U-Net was used for feature extraction from input images. The other right half-side was used for image generation.
Figure 2. The discriminator with the left half-side of the U-Net architecture. Features extracted from images were fed to the mean squared error (MSE) loss function for comparison.
Figure 3. Diagram of the cycleGAN model. In forward generation, *PETFBB images were put into the generator and **PETGE_FDG images were generated. The discriminator compared these PETGE_FDG images with ground-truth PETFDG images. In backward generation, the PETFBB images were generated from PETGE_FDG images using the same generator and discriminator.
Figure 4. Diagram of the pix2pix model. **PETGE_FDG images were generated from *PETFBB images using the one-way process, the same forward generation of the cycleGAN model.
- It’s the purpose of Generation of 18F-FDG PET Images from Delayed-phase 18F-florbetaben PET Images Using Generative Adversarial Network, Is it intended to help doctors with diagnosis in reality or for computer-aided diagnosis? For computer-aided diagnosis, SSIM as a criterion for assessing the quality of the generated images is inadequate: SSIM places more emphasis on the evaluation of structural information and perceptual quality and is suitable for scenarios that require human eye observation. Using both SSIM and PSNR would make the conclusion more convincing.
- Thanks for the good advice. We have now added the analysis using PSNR to the
manuscript.
Abstract, Results
Results: The participant demographics (age, sex, or diagnosis) showed no statistically significant differences between the training (82 participants) and validation (28 participants) groups. The mean SSIM between the PETGE_FDG and PETRE_FDG images was 0.768±0.135 for the cycleGAN model and was 0.745±0.143 for the pix2pix model. The mearn PSNR was 32.4±9.5 and 30.7±8.0. The PETGE_FDG images of the cycleGAN model showed statistically higher mean SSIM than those of the pix2pix model (p < 0.001). The mean PSNR was also higher in the PETGE_FDG images of the cycleGAN model than those of pix2pix model (p < 0.001).
Page 7, Methods and Methods, Statistical analysis, 2nd paragraph
The PSNR between the images was also measured. The PSNR is computed using the following formula:
Mean squared error (MSE) =
PSNR (dB) = 20•log10(MAXI) − 10•log10 (MSE)
Where ‘m’ and ‘n’ represent the width and height of an image, respectively, and ‘K’ represents the noisy approximation of the image. The term ‘MAXI’ refers to the maximum possible pixel value of the image. A higher PSNR value indicated that the images were more similar.
Page 9, Results, Differences in SSIM and PSNR values between PETRE-FDG and PETGE-FDG images
The mean PSNR (PSNRmean) between the PETRE-FDG and PETGE-FDG images was 32.4 for the cycleGAN and 30.7 for the pix2pix models (Table 3). The PSNR values of the cycleGAN model were significantly higher than those of the pix2pix model (p < 0.001). The PSNR values for the cycleGAN and pix2pix models are shown in the box plots provided in Figure 6.
Page 11, Discussion, 6 paragraph
SSIM and PSNR was used to compare PETGE_FDG and PETRE_FDG. SSIM was developed to predict the perceived quality of digital images and measure the similarity between two images [18]. Since then, SSIM has been widely used for image comparison by detecting perceived structural changes during image processing. Additionally, the cycleGAN model showed higher the PSNR, which is a quantitative measure of image denoising quality. SSIM aligns more closely with the human visual perception of image quality compared with PSNR [44]. Although higher SSIM and PSNR were not always equal to higher visual quality, PETGE_FDG using the cycleGAN model was statistically closer to PETRE_FDG image than PETGE_FDG image using the pix2pix model.
Figure 6. Difference in mean peak signal-to-noise ratio (PSNR) values between the cycleGAN (A) and pix2pix (B) models. The cycleGAN model shows a significantly higher PSNR than the pix2pix model.
- I could not find a paragraph in reference [5] that matches the meaning of the quoted sentence.
- We now changed the reference to a more relevant one.
Ref [5] Vandenberghe, R.; Adamczuk, K.; Dupont, P.; Laere, K.V.; Chetelat, G. Amyloid PET in clinical practice: Its place in the multidimensional space of Alzheimer's disease. Neuroimage Clin. 2013, 2, 497-511.
- The tracers used in reference [8,9] and reference [10] are PIB and florbetapir, respectively, which are not equivalent to florbetaben.
- We have now added the references [10] to a reference involving FBB
Ref [10] Tiepolt, S.; Hesse, S.; Patt, M.; Luthardt, J.; Schroeter, M.L.; Hoffmann, K.T.; Weise, D.; Gertz, H.J.; Sabri, O.; Barthel, H. Early [(18)F]florbetaben and [(11)C]PiB PET images are a surrogate biomarker of neuronal injury in Alzheimer's disease. Eur. J. Nucl. Med. Mol. Imaging 2016, 43, 1700-1709.
- It would be clearer to describe image-generation model architecture and the structure of CycleGAN/pix2pix models in diagrams instead of unclear text description.
- Thanks for the good advice. We have now added figures (Fig 1, 2, 3, and 4) to describe the image-generation model architecture and the structure of CycleGAN/pix2pix models.
Figure Legends
Figure 1. The generator with the modified U-Net architecture. The input/output image format was 64 (width) × 64 (height) × 1 (channel). After the vertical median line was virtually drawn on the U-Net diagram, the left half-side of the U-Net was used for feature extraction from input images. The other right half-side was used for image generation.
Figure 2. The discriminator with the left half-side of the U-Net architecture. Features extracted from images were fed to the mean squared error (MSE) loss function for comparison.
Figure 3. Diagram of the cycleGAN model. In forward generation, *PETFBB images were put into the generator and **PETGE_FDG images were generated. The discriminator compared these PETGE_FDG images with ground-truth PETFDG images. In backward generation, the PETFBB images were generated from PETGE_FDG images using the same generator and discriminator.
Figure 4. Diagram of the pix2pix model. **PETGE_FDG images were generated from *PETFBB images using the one-way process, the same forward generation of the cycleGAN model.
- The article does not mention the connection between the two modalities and the advantages and disadvantages of applying each to diagnosis.
- We have added a more detailed description of the connection between the two modalities and the advantages and disadvantages of applying each to diagnosis.
Page 9, Discussion, 2nd and 3rd paragrahs
The brain utilizes approximately a quarter of the body’s glucose on a daily base. Glucose is transported from the blood to the brain cells via glucose transporters. FDG, which is a glucose analog, is transported to brain cells via the same pathway but undergoes phosphorylation within the cell, which prevents it from being released from the cell. FDG accumulates without further glycolysis and is a good reflection of glucose uptake in brain cells [7]. Stimulation of neurons has been reported to coincide with FDG uptake at neuronal terminals, indicating that FDG uptake in the brain reflects neuronal activity [7]. Thus, PETFDG imaging serves as a functional imaging biomarker for assessing regional brain dysfunction caused by neuronal injury in AD. AD is characterized by decreased glucose metabolism in the posterior cingulate cortex, precuneus, and parieto-temporal cortex on PETFDG image, and in advanced cases, the decreased FDG uptake may extend to the frontal cortex. It is a non-invasive imaging test that is useful for the evaluation of disease extent in AD as well as for the differential diagnosis of other types of dementia, in which cases amyloid PET imaging may have a limited role [5,26-29]. It can also be used to predict the progression from mild cognitive impairment to AD [30] and to classify subtypes of AD [31]. However, decreased FDG uptake in the brain on PETFDG imaging is indicative of neurodegeneration. Thus, PETFDG imaging is not an appropriate imaging test for the early diagnosis of AD [32].
Early phase 11C-Pittsburgh compound B (PiB) PET (PETPiB) imaging, which is obtained within the first few minutes after PiB injection, reflects the cerebral blood pool due to the lipophilic nature and high extraction fraction of PiB [33]. The close relationship between blood supply and glucose consumption in the brain has been well-documented in several studies. In regions of the brain with neuronal injury, glucose hypometabolism and hypoperfusion are often concurrent [34,35]. Decreased tracer uptake on early phase PETPiB images is reported to closely correlate with hypometabolism in PETFDG images and low mini-mental state examination scores in patients with early-stage AD. Thus, both abnormal amyloid deposition and neurodegeneration extent in the brain may be assessed with a conventional PETPiB image [8,36]. Another amyloid tracer, 18F-florbetapir, has also been reported to have a strong correlation with PETFDG in early phase 18F-florbetapir PET imaging [37]. In a recent study comparing early phase PETFBB and PETPiB images, both early phase amyloid PET imaging showed a close correlation with PETFDG imaging. Early phase PETFBB imaging showed a slightly stronger correlation, suggesting that obtaining PETFBB images at dual time points with a single radioisotope injection has the advantages of allowing for both accurate diagnosis and assessment of progression in patients with AD [10]. This suggests that early phase amyloid PET imaging could serve as a clinically viable alternative to PETFDG for assessing neuronal injury in patients with dementia.

Round 2
Reviewer 1 Report
The reviewer has identified substantial revisions in the updated version and deems it appropriate for publication.
Minor editing of the English language required
Author Response
Comments on the Quality of English Language
Point 1. Minor editing of the English language required
- Thank you for your review and comments. As required by the journal and you, the manuscript underwent grammar editing twice by a professional English editing company. To ensure you, we are attaching the certificates issued by the company.
Reviewer 2 Report
1.The explanation of the connection between PET-FBB and PET-FDG is still insufficient. In this section, it is mainly emphasized that there is a lot of connection between PET-PIB and PET-FDG, and the connection between PET-FBB and PET-FDG is only emphasized in literal terms and lacks explanation.
2.There are still some formatting issues in the article, such as the layout of charts that need to be adjusted.
Author Response
Comments and Suggestions for Authors
Point 1. The explanation of the connection between PET-FBB and PET-FDG is still insufficient. In this section, it is mainly emphasized that there is a lot of connection between PET-PIB and PET-FDG, and the connection between PET-FBB and PET-FDG is only emphasized in literal terms and lacks explanation.
- There are many articles showing the relationship between early PETFBB and PETFDG. We now emphasized the relationship between early PETFBB and PETFDG and added relevant references to the discussion section.
Discussion section page 10 1st paragraph
Before:
In a recent study comparing early phase PETFBB and PETPiB images, both early phase amyloid PET imaging showed a close correlation with PETFDG imaging. Early phase PETFBB imaging showed a slightly stronger correlation, suggesting that obtaining PETFBB images at dual time points with a single radioisotope injection has the advantages of allowing for both accurate diagnosis and assessment of progression in patients with AD [10].
After:
In recent studies regarding early phase PETFBB images, early phase PETFBB images showed a close correlation with PETFDG image [25,38,39]. In one study, early phase PETFBB images showed a slightly stronger correlation than PETPiB images, suggesting that obtaining PETFBB images at dual time points with a single radioisotope injection has the advantages of allowing for both accurate diagnosis and assessment of progression in patients with AD [10].
Point 2. There are still some formatting issues in the article, such as the layout of charts that need to be adjusted.
- Thank you for your comments. Due to the short deadline, formatting was not perfect at the time of resubmission. The manuscript will undergo further formatting under the guidance of the editorial board. The charts have also been modified to ensure a higher resolution and a good readability. We will be sure to keep close contact with the editorial board during the reformatting process. Thank you.